# Investigating the Impact of Resilience, Responsiveness, and Quality on Customer Loyalty of MSMEs: Empirical Evidence

**Nourhan Ah. Saad** [1,2,*] , **Sara Elgazzar** [1] **and Sonja Mlaker Kac** [2]

1 Logistics of International Trade Department, College of International Transport and Logistics, Arab Academy for Science, Technology and Maritime Transport, Alexandria 1209, Egypt; sara.elgazzar@aast.edu
2 Faculty of Logistics, Maribor University, 3000 Celje, Slovenia; sonja.mlaker@um.si
* Correspondence: nourhan_ahmed@aast.edu

**Abstract:** Due to the importance of the micro, small, and medium-sized enterprises (MSMEs) sector and the negative implications of COVID-19, which resulted in decreasing resource availability, shortages of supply, declining consumer demand and requirements, and a lack of consumer satisfaction and loyalty, this research investigates the impact of resilience, responsiveness, and quality on customer loyalty in MSMEs. An online questionnaire was conducted on MSMEs' end consumers in the Egyptian context. The analysis was conducted through Amos and SPSS, and the research hypotheses were tested through covariance-based structural equation modelling for 891 valid questionnaires. The findings exposed that there is a positive significant impact for operational resilience (flexibility and technology adoption), responsiveness (delivery fulfillment and speed and after-sale service), and product/service quality on customer loyalty in terms of behavioral, attitudinal dimensions. It contributes to understanding how MSMEs could enhance their sustainable performance (resilience, responsiveness, quality) to reach better customer loyalty. This research presents insights on how the MSMEs sector can adapt to the dynamic business environment in terms of COVID-19 crisis and consumer behavior, which has changed the nature and needs of the market and consumers. In addition, this research extends the theories of Resource-Based View (RBV), Dynamic Capability View (DCV), and Theory of Consumption Value (TCV) in an empirical contribution through filling the gap in understanding consumers' needs in terms of resilience, responsiveness, and quality.

**Keywords:** responsiveness; resilience; quality; Egypt; customer loyalty; micro; small and medium-sized enterprises

## 1. Introduction

The COVID-19 pandemic caused unprecedented and severe interruptions in today's business environment, which is considered the first and foremost human tragedy. Additionally, the COVID-19 pandemic has had a huge impact on countries' economic situations, where the global economy experienced the deepest economic recession since the Second World War [1]. Moreover, the pandemic has had a large influence on regional Micro-, Small-, and Medium-sized Enterprises (MSMEs) sector, and this sector occupies the worst position after the COVID-19 outbreak [2].

COVID-19 has hit the MSMEs' business, where they suffered from several challenges such as supply shortage, decline in demand, labor force reduction, and customer dissatisfaction. Moreover, the pandemic has badly harmed MSMEs' businesses, where shortage of resources and not meeting customer requirements have become the most significant outcomes caused by the pandemic [3]. Thus, the COVID-19 pandemic has had an unprecedented pressure on several industries, where they have to reorganize their operations to

ensure continuity of operations and availability of products as well as reaching consumers' demand and loyalty [4].

Customers are considered one of the most important assets for organizational survival; thus, organizations should focus on improving organizational products/services and gaining competitive advantage through meeting consumers' needs and demands [5]. Satisfying customers' needs is the main factor in growing their loyalty [6]. Customers are considered satisfied when the product/service they receive in reality exceeds their expectations [7]. Satisfied customers tend to build strong and long relationships with organizations that will nurture customer loyalty [8].

Moreover, organizations should focus more on enhancing their operations' performance to provide customers with a qualified product/service on time and in a perfect condition, and to reach a superior position compared to other competitors [9]. In today's competitive environment, customers are the decisive factor for organizational existence and development [10]. Customer satisfaction and loyalty are a significant goal for every organization, where the level of competition between organizations has become aggressive, organizations have to focus on the whole operational processes, including after-sale services [11].

Nowadays, organizations are operating in a dynamic business environment [12], where the COVID-19 pandemic has dramatically affected all businesses' operations [13] and threatened their survival [14]. Furthermore, strategies for organizational survival are based on their response to the changing environment [15]. Moreover, organizations have to become more flexible, resilient, and innovative to survive in the marketplace [12]. Resilience allows organizations to take actions and develop contingency plans to deal with unstable circumstances as a way to enhance performance of organizations [16], where scholars argue that the key ways to help organizations to deal with the changing business environment and uncertainty are innovation [17] and flexibility [18]. Firstly, flexibility refers to organizational capabilities to adapt to complex environments and tasks [19], where it is considered as a principal weapon for achieving an organizational competitive advantage in most significant uncertainties [20]. Secondly, innovation focuses on technology adoption, as it was initially developed for larger organizations [21], and where technology adoption creates potential opportunities for enhancing operations' processes for the MSMEs sector in particular [22].

Moreover, responsiveness is considered as one of the main chances for MSMEs' success and survival, especially in the context of the COVID-19 pandemic [23,24]. In addition to the main keys for enhancing the responsiveness level were speed and fulfillment of order and after-sale services, as customers nowadays expect more customized products/services and fast response to their complaints and after-sale services [25].

Additionally, quality is one the main keys for enhancing the operational performance of organizations among products/services by taking consumers' requirements and needs into consideration to improve the overall performance and customer satisfaction level [26]. Due to dynamic changes in consumers' demands, quality has become one of the top priorities of organizations to gain a better competitive advantage [27].

In the context of organizational survival and customer loyalty, organizations should focus on understanding consumers' behavior in choosing products/services through empirically extending the Theory of Consumption Value (TCV) [28]. Furthermore, organizations should adapt quickly to the changing market environment by using their resources wisely through integration of Dynamic Capability View (DCV) theory and Resource-Based View (RBV) theory [29]. Therefore, this research extends the theoretical foundations of the three theories through empirical evidence on the MSMEs sector in developing countries, particularly the Egyptian context.

In Egypt, Micro, Small and Medium-sized Enterprises (MSMEs) have high potential for growth and development and their employment contribution [30]. Moreover, MSMEs sector plays a vital role in economies, where this sector is the most attractive and remarkable in the world's economic growth and development [31]. Additionally, MSMEs sector represents more than 90 percent of all businesses and 75 percent of national value-added activities [32].

Moreover, the MSMEs sector plays an integral role in economic growth and development in developing countries and has dramatically suffered from uncertainty and implications of COVID-19 outbreaks [33]. Furthermore, previous literature and empirical studies are still limited for the MSMEs sector compared to large enterprises [34], especially in developing countries [35].

This research contributes in several ways: first, it explores the important aspects for enhancing operational performance in terms of resilience, responsiveness, and quality and investigates their influence on customer loyalty in terms of their behavioral and attitudinal loyalty; secondly, this research focuses on MSMEs' manufacturing and service sectors in emerging markets, in particular in the Egyptian context; thirdly, this research empirically extends the theoretical foundations of TCV, RBV and DCV.

Therefore, this research aims to investigate the impact of resilience, responsiveness, and quality on MSMEs' consumer loyalty in the Egyptian context. This research is organized as follows: the following section reviews conceptual model and hypotheses development for the research model; Section 3 outlines the methodological tools and methods applied in this research, while Section 4 presents the research analysis and findings. Section 5 represents discussion of research, and finally conclusion and recommendations for future studies are presented in Section 6.

## 2. Literature Review and Hypotheses Development

### 2.1. Operational Resilience and Customer Loyalty

Business and organizational resilience have become an interesting area of research for academia and practitioners [36,37], where resilience is known as the organizational capabilities necessary to cope with disruption and retaining the same businesses' functions and structure [38], as well as their ability to return stronger after disturbance occurrence [37].

Moreover, COVID-19 created a severe shock on both the supply and demand sides due to unprecedented disruptions [39,40], in addition to the organizational ability to cope with constant changes that occur in customers' needs, demands, and expectations, materials and manpower shortage, and organizational capabilities used to cope with pandemic lockdowns [41]. Therefore, resilience has become an important aspect to enhance the performance of organizations and to respond to unexpected shocks [42].

Due to the era of the COVID-19 pandemic, organizations must make decisions quickly to cope with uncertainties, especially in emerging markets [43]. Organizations should adopt resilience in order to optimize their organizational flow [44]. Additionally, resilience helps organizations to better calibrate their overall performance during unexpected events and thus cope with the dynamic environment and the changing consumer behaviors [45]. Furthermore, previous studies indicate that organizations that adopt a resilient strategy have the mechanisms needed to deal with disruptions and enable them to reach superior performance outcomes [46–48].

Furthermore, a resilient organization should focus on understanding the full situation and challenging itself to continually improve its products/services, improve its overall organizational performance, grow organizational sustainability, and enhance the level of customers' satisfaction and loyalty [49]. Therefore, organizations should focus more on enhancing resilience practices to enhance their overall performance and improve customer loyalty level, especially after the COVID-19 outbreak negatively affected most of the MSMEs [50]. In order to overcome operational and Supply Chain (SC) problems, organizations should increase their reactive response and enhance resilience through flexibility practices and innovation adoption [51].

Flexibility is considered as the organizational ability to operate in a more turbulent environment [52]. It is possible to consider the operational flexibility as one of the main drivers to enhance organizational performance on an operational level, where it refers to the organizational ability to reconfigure organizational resources to offer adequate products/services in the dynamic and fluctuated market environment [53,54].



While technology adoption has become a crucial aspect for organizational success and market competitiveness in the era of internationalizing operations and globalization [55], technological innovation has a crucial role in improving performance of MSMEs and achieving sustainable growth and competitiveness [56]. Moreover, technology adoption has become an essential tool for leveraging competitive advantage and ensuring interaction with customers [57]. Therefore, flexibility and innovation are considered crucial aspects for improving organizational daily operation as well as helping organizations to mitigate the negative impact of uncertainty disruption [58].

Most of the previous literature investigated the positive significant impact of resilience on overall organizational performance and survival, especially after the occurrence of the COVID-19 pandemic [59]. However, limited studies investigated the impact of resilience on levels of customer loyalty [60]. Additionally, [61] they confirmed the positive impact of flexibility on levels of customer loyalty and [62] confirmed the positive impact of technology adoption (artificial intelligence) on customer loyalty.

Finally, resilience enables organizations to respond to the changing demand in a dynamic environment and cope with sudden disturbance, where resilience helps organizations to improve operational efficiency and performance and meet customers' needs and loyalty [63], in addition to the positive impact of resilience on performance [64], which leads to a better level of customer loyalty [65]. Based upon the previous discussion, the following hypothesis could be proposed:

**Hypothesis 1 (H1).** *There is a positive impact of resilience on customer loyalty.*

### 2.2. Responsiveness and Customer Loyalty

Organizational responsiveness refers to the organizational ability to respond immediately to any environmental changes that may affect business [66]. Organizational responsiveness is considered one of the most significant organizational capabilities to directly produce organizational competitive advantage, which relies on the organizational ability to respond quickly to any environmental changes and leads to customer retention and generates value for customers [67]. Organizational responsiveness is known as a combination of a unique set of skills, routines, and processes to help organizations to respond quickly to any external changes that occur in the marketplace as well as to meet customers' requirements in an efficient and effective manner [68]. Thus, organizational responsiveness is considered as a mechanism with which to link both external capabilities and internal behaviors to support the operations department to react successfully to consumers' needs [67].

Furthermore, operational responsiveness can be defined as the organizational operations' ability to respond quickly to market changes, including consumers' demand [69], where organizations have to react in an effective way towards consumers' demand and competitor-related changes [70]. Thus, organizations should fulfill consumers' demands in an efficient and effective way and to deliver them as fast as possible to avoid consumers' disappointment or dissatisfaction [71], where delivery fulfillment and speed refer to the organizational ability to deliver products/services in a quick and reliable way to the customer [72].

For organizations to have more responsiveness, they should focus on fulfilling consumers' needs and demands in the fastest manner, where order/delivery fulfillment refers to organizational ability to deliver consumers' orders/requirements in a perfect condition and on time [73]. Moreover, organizations should focus on fulfilling consumers' orders correctly and deliver the required items as quickly as possible, and thus organizations will be as responsive as possible to consumer inquires within a reasonable timeframe. Furthermore, customer service and after-sale services are considered the key element for achieving more responsive results [74].

Many organizations focus mainly on consumers' requirements as consumers are considered the main assets for organizational survival, where organizations focus on providing responsiveness through practices of fast and reliable delivery [75], where quick delivery

and adequate return policies are expected to significantly affect consumers' behavior in a positive way and thus enhance their level of satisfaction and loyalty [76]. Moreover, order fulfillment incorporates all organizational activities that are necessary to deliver products/services to the consumers in a successful way, including order fulfillment and speed as well as after-sale services, which include returns handling practices and policies [77].

In the mobile telecommunication sector, [78] illustrated that there is no significant relationship between responsiveness and customer loyalty. Meanwhile, [79] confirmed the positive impact of responsiveness on customer loyalty in the coffee shop industry. Furthermore, [80] confirmed the positive influence of responsiveness on customer loyalty in telecommunication industries. Finally, in the banking sector, [81] confirmed the positive impact of responsiveness on customer loyalty.

Additionally, [82] empirically investigated the positive significant impact of responsiveness on customer loyalty in m-commerce applications. The authors of [83] confirmed the positive influence of responsiveness on customer loyalty in mobile shopping. Additionally, [84] exposed the positive impact of responsiveness on customer loyalty in restaurants. Furthermore, previous studies confirmed the positive significant effect of delivery fulfillment [85] and speed [71] on customer loyalty. Additionally, [86,87] confirmed the positive significant influence of after-sale services on overall customer loyalty.

Nowadays, consumers focus more on personalized products/services as well as responding faster to their complaints [25]. Organization should focus on the provision of shipping information, service breakdowns, order delays, return items or refund requests to enhance the responsiveness level of organizations, and thus consumers will not switch to other competitors and will increase their intention to re-purchase from the same organization as they deliver to them the promised product/service [74,88]. Based upon the previous discussion, the following hypothesis could be proposed:

**Hypothesis 2 (H2).** *There is a positive impact of responsiveness on customer loyalty.*

### 2.3. Quality and Customer Loyalty

Operational performance is considered as the organizational ability to provide a product/service which is consistently correct, provide delivery on-time, operational flexibility where appropriate, along with cost effectiveness [89]. Moreover, operational performance is considered as a multidimensional construct including quality, delivery, flexibility, and cost [90,91], where quality is a key element for improving operational performance through continually enhancing product/service quality to meet and exceed customers' expectations, which has resulted in improving customer loyalty [92].

Product/service quality is widely acknowledged as a key driver for enhancing operational performance of organizations and gaining competitive advantage [93]. Moreover, quality is considered as a main determinant for enhancing operational performance [94].

In the era of globalization, organizations should improve their overall performance and customer loyalty level through improving quality of products/services, which is conducted through reaching and fulfilling customers' needs and demands as well as exceeding their expectations [95,96]. Furthermore, quality is considered the key aspect for organizational survival in the global market [97].

Previous studies confirmed the positive impact of product/service quality on customer loyalty in the manufacturing sector [98], while [99] also confirmed the positive influence of product/service quality on customer loyalty in the banking sector. Additionally, [100] confirmed the positive impact of product/service quality on customer loyalty in the fashion sector. Finally, [101] confirmed the positive impact of quality of products/services on customer loyalty in cosmetics industry.

Moreover, previous studies identified that enhancing quality of products/services will lead to better customer satisfaction and loyalty, which resulted in re-purchasing intention [98]. Therefore, organizations should focus on enhancing the product quality to reach better customer loyalty as investigated by [102] in the fast-food industry and

enhancing the quality of services to increase customer loyalty level as investigated by [103] in the banking sector. Thus, the better quality an organization provides to customers, the better customer loyalty they will experience. Based upon the previous discussion, the following hypothesis could be proposed:

**Hypothesis 3 (H3).** *There is a positive impact of product/service quality on customer loyalty.*

Based on the previous discussion, the hypothesized framework is illustrated in the following figure (Figure 1).

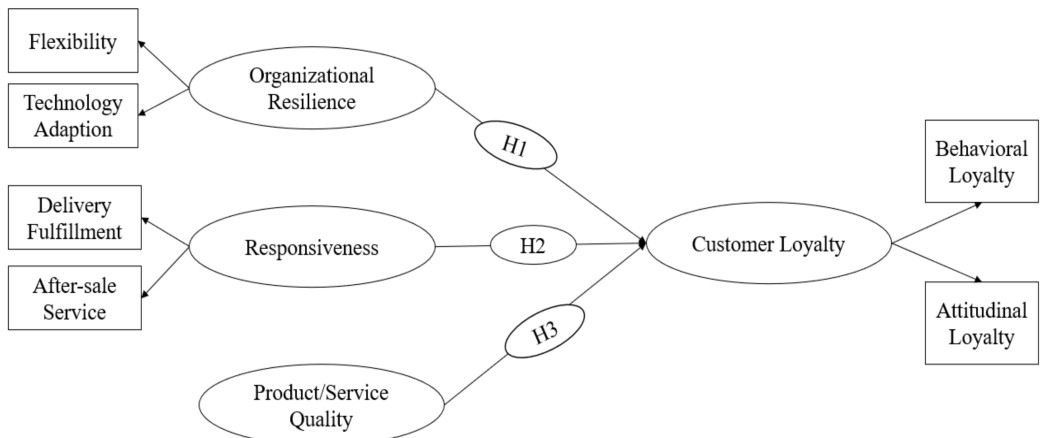

**Figure 1.** Research model and proposed relationships.

The hypothesized framework illustrated in the previous figure is conceptualized to help organizations to manage their internal resources and deal with an external dynamic market environment, especially after implications of COVID-19 outbreaks, where RBV theory refers to the organizational capabilities to manage their internal resources wisely through grabbing their competitive advantage through integration between their internal resources such as: quality of their products/services and their responsiveness to customers in terms of (delivery fulfillment and speed, after-sale services), as the main assumption for this theory is that organizational resources must be immobile and heterogenous [104]. Moreover, DCV theory refers to the organizational ability to cope with a dynamic market environment through dealing with constant environments such as resilience in terms of flexibility and technology adoption, which are considered main practices due to the implications of COVID-19 [105]. Additionally, TCV theory refers to the consumers' preferences according to which they prefer a specific product/service from specific organization, which leads to understanding how to enhance consumers' loyalty, which includes their behavioral and attitudinal loyalty dimensions [106].

The collaboration between RBV and DCV theories helps organizations to use their resources in an efficient way and cope with a changing market environment [107], through the adoption of several practices such as resilience (flexibility and technology adoption) to cope with turbulence situations [108]. Moreover, organizations should manage their resources wisely through adopting responsiveness practices (delivery fulfillment and speed and after-sale services), and providing products/services with the highest quality, in addition to the merging of previous theories with TCV to focus on consumers' behavior and repurchase intention in terms of customer loyalty (behavioral and attitudinal dimensions) towards products/services [109]. Therefore, this research focused on empirically investigating the hypothesized framework.

## 3. Materials and Methods

The aim of this research is to investigate the impact of resilience, responsiveness, and quality on customer loyalty in the Egyptian context with empirical evidence on the

MSMEs sector. To investigate the relationship between research variables, a theoretical framework was empirically tested through 891 surveys distributed to end-consumers. The following sub-sections illustrate in detail the methodological approach.

Data were gathered and analyzed through SPSS version 23 and AMOS version 21. Firstly, SPSS was used to analyze the descriptive statistics data for demographic characteristics and multi-collinearity analysis, while AMOS is used for analyzing both the pilot and main study to assess reliability and validity through Confirmatory Factor Analysis (CFA), model fit, and testing research hypotheses.

### 3.1. Sampling and Data Collection

Egypt is in the process of restructuring a competitive framework for the MSMEs and entrepreneurship sector, where the MSMEs sector represents more than 90 percent of all businesses, almost 60 percent of the employment rate, and around 75 percent of the national value-added activities [32]. Thus, this research uses data on emerging markets, particularly Egyptian MSMEs sector, where data were collected from end-consumers who dealt frequently with any MSMEs (i.e., manufacturing, sector sectors).

The hypothetical framework was validated and tested through a self-administered questionnaire, where data were collected from November 2021 to February 2022. A snowball sampling technique was used in this research for both pilot and main studies, as the snowball sampling technique is considered the most appropriate sampling method when the research population is unknown and there is no access availability to these data [110]. Moreover, CFA and Covariance-Based Structural Equation Modelling (CB-SEM) methods were used to analyze the collected data. CFA and CB-SEM accepted an adequate sample size with a minimum of 200 valid responses [111,112]. Online surveys were distributed to 250 respondents for the pilot study and 1000 for the main study. Firstly, for the pilot, a total of 240 respondents replied with 16 incomplete surveys and 224 valid ones, with a response rate of 89.6%. Secondly, for the main study, a total of 910 respondents replied with 19 incomplete surveys and 891 valid ones, with a response rate of 89.1%. The characteristics of research sample for the main study in terms of gender, age, educational level, income, employment status, and organizational sectors are illustrated in Table 2.

### 3.2. Survey Instrument and Design

The questionnaire conducted in this research was adapted from previous studies in order to capture the causal relationship between variables included in this study, where the data collected were through a structured survey. The structured survey is divided into three sections: the first section identifies demographic characteristics of respondents as illustrated in Table 2, the second section illustrates resilience in terms of flexibility and technology adoption, responsiveness in terms of delivery fulfilment and speed and after-sale services, and product/service quality, and the third section identifies customer loyalty in terms of behavioral loyalty dimension and attitudinal loyalty dimension.

The measurement items of resilience were measured as a second-order factor that consists of two first-order factors: flexibility that includes six items used by [113] and technology adoption that includes six items used by [114]. Moreover, the measurement items of responsiveness were measured as a second-order factor that consists of two first-order factors: delivery fulfilment and speed that include five items adopted by [73] and after-sale services that include six items adopted by [115]. Furthermore, the measurement items of product/service quality were measured through three items adapted by [116].

Finally, the measurement items of customer loyalty were measured as a second-order factor that consists of two-first order factors: behavioral loyalty and attitudinal loyalty dimensions, which were adopted from [117] as all first-order factors were measured using three measurement items.

All variables were assessed through a five-point Likert scale (1 = totally disagree, 2 = disagree, 3 = average, 4 = agree, 5 = totally agree). Furthermore, the survey was translated from English to Arabic, and then a back translation process method was carried

out [118]. Finally, the questionnaire was pretested through presenting it to industry experts and academics to ensure content validity of the survey [119].

### 3.3. Pilot and Pre-Testing

A pilot testing was conducted using 224 valid surveys, including 90 manufacturing companies and 134 service companies. To assess the relatability and validity of collected data, a Confirmatory Factor Analysis (CFA) was performed using AMOS 21.

Confirmatory Factor Analysis (CFA) was used to assess the reliability and validity of the collected data [120,121]. According to the CFA, all items with standardized loadings less than 0.4 were dropped [122]. The researcher dropped six statements from the resilience construct: "organization changes production planning quickly", "organization operates in volatile markets", "organization introduces new products/services on ongoing basis", "organization should provide inadequate budget to invest in new technologies", "organization should consider the technology as a main driver for business growth", and "technology allows organization to accomplish specific tasks more quickly", while five statements were dropped from the responsiveness construct: "organization provides product/service that meet customer expectations", "organization provides on-time delivery", "responsiveness to handle customer complaints", "organization provides maintenance and repair support for defects", and "organization provides online/telephone service support", whereby one statement was dropped from the behavioral loyalty dimension: "if I had it to do all over again, I would buy products/services from this organization", and finally, one statement was dropped from the attitudinal loyalty dimension: "I would say positive things about the organization's products/services to other people". After dropping items, the Average Variance Extracted (AVE) and Composite Reliability (CR) were improved, where AVE should exceed 0.5 and CR should exceed 0.7 [123]. The following table (Table 1) illustrates the CFA for pilot analysis.

**Table 1.** Pilot analysis based on confirmatory factor analysis.

| Construct | Items | Standardized Loadings | CR | AVE |
|-----------|-------|----------------------|-----|-----|
| **Flexibility** | FLX1 | 0.987 | 0.783 | 0.561 |
| | FLX2 | 0.599 | | |
| | FLX3 | 0.592 | | |
| **Technology Adoption** | TA1 | 0.862 | 0.810 | 0.593 |
| | TA2 | 0.826 | | |
| | TA3 | 0.594 | | |
| **Delivery Speed and Fulfilment** | DSF1 | 0.631 | 0.805 | 0.583 |
| | DSF2 | 0.787 | | |
| | DFS3 | 0.856 | | |
| **After-Sale Service** | A_SS1 | 0.747 | 0.824 | 0.611 |
| | A_SS2 | 0.705 | | |
| | A_SS3 | 0.882 | | |
| **Product/Service Quality** | PSQ1 | 0.633 | 0.806 | 0.585 |
| | PSQ2 | 0.868 | | |
| | PSQ3 | 0.775 | | |
| **Behavioral Loyalty** | BLD1 | 0.879 | 0.768 | 0.623 |
| | BLD2 | 0.796 | | |
| **Attitudinal Loyalty** | ALD1 | 0.953 | 0.949 | 0.903 |
| | ALD2 | 0.948 | | |

After conducting CFA, the model fit indices were assessed the final research model through Amos and confirmed the overall fitness of the model, where Chi-square ($x^2$) = 397.095, $p$-value < 0.01, degrees of freedom (df) = 129, $x^2$/df = 3.078, GFI = 0.849, IFI = 0.914, and CFI = 0.913 [124].

## 4. Research Analysis and Findings

Research hypotheses were tested through Covariance-Based Structural Equation Modelling (CB-SEM), as the main purpose of this research is to test and confirm existing theory and hypotheses development rather than predicting and developing theory as this study contains an appropriate/large sample size to empirically test data [125]. The data were first tested through a single factor test to ensure if there is any common method bias [126], where the results exposed that no single factor exceeded 50 percent in variance explanation [127]. Then, non-response bias was tested to ensure that there is no difference between early and late responses, where the results confirmed that there is no non-response bias as the *p*-values were greater than 0.05 [128].

The following sections will analyze demographic characteristics, measurement and structural models will be tested, and finally the hypothetical framework will be analyzed.

### 4.1. Descriptive Statistics

A total of 891 respondents were analyzed to test the hypothetical model. The demographic characteristics of respondents were as follows: 424 male and 467 female; 408 respondents received a monthly income less than EGP 5000, while 209 received monthly income ranging from EGP 5001 to 10,000, whereas 134 received monthly income ranging from EGP 10,001 to 15,000, and 140 respondents received monthly income more than EGP 15,000. Moreover, 380 respondents were undergraduate students in school or college and 511 respondents were postgraduates whether bachelor's degree holders, master's degree holders or philosophy doctorate holders. Furthermore, 402 of the respondents ranged from 16 to 25 years; 242 ranged from 26 to 35 years; 136 ranged from 36 to 45 years; and 111 were above 45 years old. Additionally, 612 respondents were single and 279 were married. Additionally, 291 respondents were formally employed, while 275 were self-employed, 309 were students, and 16 were unemployed. Finally, 435 respondents dealt with manufacturing/product enterprise, and 456 respondents dealt with service enterprise. Detailed information of respondents' characteristics is illustrated in the following table (Table 2).

**Table 2.** Demographic information for respondents.

| Characteristics | Sub-Characteristics | Frequency | Percentage |
|---|---|---|---|
| Gender | Male | 424 | 47.6% |
| | Female | 467 | 52.4% |
| Monthly Income (EGP) | >=5000 | 408 | 45.8% |
| | 5001–10,000 | 209 | 23.5% |
| | 10,001–15,000 | 134 | 15.0% |
| | >15,000 | 140 | 15.7% |
| Educational Level | Undergraduate | 380 | 42.7% |
| | Postgraduate | 511 | 57.3% |
| Age | 16–25 | 402 | 45.1% |
| | 26–35 | 242 | 27.2% |
| | 36–45 | 136 | 15.3% |
| | >45 | 111 | 12.5% |
| Marital Status | Single | 612 | 68.7% |
| | Married | 279 | 31.3% |
| Employment Status | Formally Employed | 291 | 32.7% |
| | Self Employed | 275 | 30.9% |
| | Student | 309 | 34.7% |
| | Unemployed | 16 | 1.8% |
| Sector | Manufacturing/Product | 435 | 48.8% |
| | Service | 456 | 51.2% |

### 4.2. Measurement Modeling Testing

Reliability and validity assessments of the measurement model were carried out through CFA. Reliability analysis was assessed through the value of *CR* that was calculated through the following rule ($CR = \frac{(\sum \Lambda)^2}{(\sum \Lambda)^2 + \sum(1-\Lambda^2)}$), where $\Lambda$ represents the standardized loading. The results of reliability analysis exposed that all the research's constructs are more than 0.7. Then, the research model was validated through standardized regression weights, where all constructs' items exceeded 0.4. Additionally, AVE was calculated for each construct through the following rule ($AVE = \frac{\sum \Lambda^2}{n}$), where $\Lambda$ represents the standardized loading and n represents the number of items. The results indicated that all research constructs have *AVE* more than 0.5. The following Table (Table 3) illustrated the results of reliability and convergent validity for the research constructs.

**Table 3.** Reliability and convergent validity for the main study.

| Construct | Items | Standardized Loadings | CR | AVE |
|---|---|---|---|---|
| Flexibility | FLX1 | 0.792 | 0.780 | 0.543 |
| | FLX2 | 0.737 | | |
| | FLX3 | 0.677 | | |
| Technology Adoption | TA1 | 0.765 | 0.766 | 0.523 |
| | TA2 | 0.715 | | |
| | TA3 | 0.687 | | |
| Delivery Fulfilment and Speed | DFS1 | 0.717 | 0.788 | 0.554 |
| | DFS2 | 0.766 | | |
| | DFS3 | 0.749 | | |
| After-sale Service | A_SS1 | 0.689 | 0.762 | 0.517 |
| | A_SS2 | 0.692 | | |
| | A_SS3 | 0.773 | | |
| Product/Service Quality | PSQ1 | 0.651 | 0.771 | 0.531 |
| | PSQ2 | 0.798 | | |
| | PSQ3 | 0.729 | | |
| Behavioral Loyalty Dimension | BLD1 | 0.791 | 0.739 | 0.586 |
| | BLD2 | 0.739 | | |
| Attitudinal Loyalty Dimension | ALD1 | 0.765 | 0.713 | 0.555 |
| | ALD2 | 0.724 | | |

In the previous table, the results identified that all constructs' items reached statistical acceptable levels, where convergent validity and reliability analysis reached an adequate level. The discriminant validity will be assessed through computing the square root of AVE and compare it with the correlations between research variables as recommended by [129], which is illustrated in Table 4.

**Table 4.** Discriminant validity for the main study.

| | Cronbach's Alpha (α) | AVE | SQR (AVE) | FLX | TA | DFS | A_SS | PSQ | BLD | ALD |
|---|---|---|---|---|---|---|---|---|---|---|
| **FLX** | 0.773 | 0.543 | 0.737 | | | | | | | |
| **TA** | 0.764 | 0.523 | 0.723 | 0.489 ** | | | | | | |
| **DFS** | 0.768 | 0.554 | 0.744 | 0.552 ** | 0.587 ** | | | | | |

**Table 4.** *Cont.*

| | Cronbach's Alpha (α) | AVE | SQR (AVE) | FLX | TA | DFS | A_SS | PSQ | BLD | ALD |
|---|---|---|---|---|---|---|---|---|---|---|
| **A_SS** | 0.744 | 0.517 | 0.719 | 0.542 ** | 0.489 ** | 0.662 ** | | | | |
| **PSQ** | 0.786 | 0.531 | 0.728 | 0.543 ** | 0.491 ** | 0.524 ** | 0.522 ** | | | |
| **BLD** | 0.729 | 0.586 | 0.765 | 0.453 ** | 0.376 ** | 0.426 ** | 0.404 ** | 0.491 ** | | |
| **ALD** | 0.717 | 0.555 | 0.745 | 0.379 ** | 0.432 ** | 0.473 ** | 0.497 ** | 0.461 ** | 0.564 ** | |

Notes: ** Correlation is significant at the 0.01 level (2-tailed); AVE = Average Variance Extracted, FLX = Flexibility, TA = Technology Adoption, DFS = Delivery Fulfilment and Speed, A_SS = After-Sale Services, PSQ = Product/Service Quality, BLD = Behavioral Loyalty Dimension, ALD = Attitudinal Loyalty Dimension.

Moreover, a new method has emerged to accurately test discriminate validity, which is the Heterotrait–Monotrait ratio of correlation (HTMT) methods which was developed by [130] and is illustrated in the following table (Table 5).

**Table 5.** Heterotrait–Monotrait discriminant validity of the main study.

| | FLX | TA | DFS | A_SS | PSQ | BLD | ALD |
|---|---|---|---|---|---|---|---|
| FLX | | | | | | | |
| TA | 0.635 | | | | | | |
| DFS | 0.743 | 0.710 | | | | | |
| A_SS | 0.719 | 0.687 | 0.881 | | | | |
| PSQ | 0.663 | 0.633 | 0.708 | 0.685 | | | |
| BLD | 0.546 | 0.521 | 0.614 | 0.593 | 0.630 | | |
| ALD | 0.574 | 0.589 | 0.647 | 0.626 | 0.646 | 0.771 | |

The results shown in the previous table confirmed that discriminant validity of the research model reached an adequately acceptable level, where all correlations are less than 0.9 [131].

Moreover, model fit indices were assessed through Amos to confirm that the research model has an acceptable level of goodness of fit, where the following table (Table 6) illustrates the model fit indices along with the threshold of the measurement model.

**Table 6.** Model fit indices along with threshold values.

| Measure Indices | Results | Threshold | Threshold's Reference |
|---|---|---|---|
| Chi-square | 757.538 | | |
| *p*-Value | <0.01 | | |
| Degree of Freedom | 127 | | |
| CMIN/df | 5.965 | <2 (excellent) <3 (good) <5 (reasonable fit) | [131,132] |
| GFI | 0.917 | >0.90 (excellent) >0.80 (acceptable) | [125] |
| AGFI | 0.876 | >0.90 (excellent) >0.80 (acceptable) | [131] |
| NFI | 0.901 | >0.90 | [133] |
| CFI | 0.916 | >0.90 | [134] |
| IFI | 0.916 | >0.90 | [126] |
| RMR | 0.029 | <0.09 | [131] |
| RMSEA | 0.075 | <0.05 (good) 0.05–0.10 (moderate) >0.10 (bad) | [131] |

The results illustrated in the previous table confirmed the goodness of fit indices for the measurement model, however CMIN/df is greater than the accepted level and could

be ignored as it is directly affected by large sample size, which is clearly presented in this research (*n* = 891) [135].

Moreover, the inter-correlation between research variables will be analyzed through a multi-collinearity test, where a tolerance value for resilience, responsiveness, and product/service quality are 0.462, 0.485, and 0.591, respectively, and the values for the variance inflation factor (VIF) for resilience, responsiveness, and product/service quality are 2.164, 2.061, and 1.693, respectively. Therefore, the results indicated that there is no multi-collinearity problem between research variables as the tolerance values were greater than 0.1 and VIF values were less than 3.3 [136,137].

*4.3. Hypotheses Testing*

Based on the analysis and results of the measurement and structural models, which were estimated using Amos 21.0, the hypothesized framework shown in Figure 1 is considered an adequate representation for the entire set of causal relationships.

Figure 2 illustrates the estimated path coefficient and significance level along with a structural model, where the results showed that there was a significant direct impact of resilience (i.e., flexibility and technology adoption) on customer loyalty (β = 0.132 and *p*-value < 0.05). Moreover, the results confirmed that there is a positive significant impact responsiveness (i.e., delivery fulfilment and speed and after-sale service) on customer loyalty (β = 0.378 and *p*-value < 0.05). Finally, results confirmed a significant positive impact product/service quality on customer loyalty (β = 0.418 and *p*-value < 0.05); therefore, H1, H2, and H3 are supported.

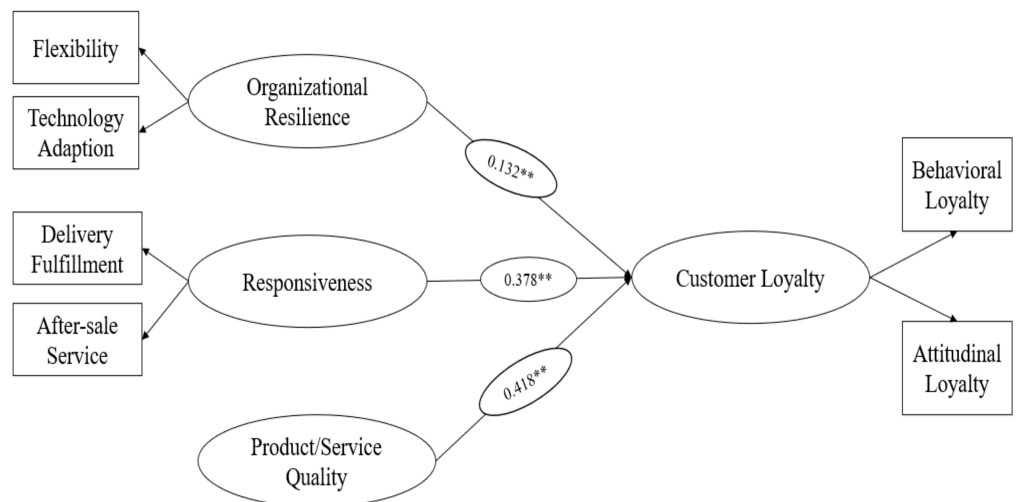

**Figure 2.** Research model results. Note: ** Significant level at 0.01.

Based on the previous discussion, it was concluded that H1, H2, and H3 are accepted for the whole entire sample. The following table (Table 7) summarizes research hypotheses' testing.

**Table 7.** Research hypotheses testing.

| Path | Hypothesis Testing |
| --- | --- |
| Resilience → customer loyalty | H1—accepted |
| Responsiveness → customer loyalty | H2—accepted |
| Product/service quality → customer loyalty | H3—accepted |

## 5. Research Discussion

This research focused on investigating the impact of resilience, responsiveness, and product/service quality on customer loyalty, and the results were empirically investigated

on the Egyptian MSMEs sector. The findings of this research help the MSMEs sector to use their resources efficiently to cope with the changing environment and the consumers' changeable demands to reach a better customer loyalty level.

Firstly, the positive and significant influence illustrated of resilience (flexibility and technology adoption) on customer loyalty (behavioral and attitudinal loyalty) was supported [59,60], where authors exposed that there was a positive impact of resilience and sustainability practices on customer loyalty. Moreover, [61] empirically tested whether flexibility helps service companies to enhance their customer loyalty level. Additionally, [138] tested the positive impact of technology adoption on customer loyalty. Therefore, MSMEs sector should enhance their resilience practices according to DCV theory to enhance consumers' repurchase intention as illustrated in TCV and thus enhance the overall level of consumers' loyalty.

Secondly, the illustrated positive and significant influence of responsiveness (delivery fulfillment and speed and after-sale service) on customer loyalty was supported by [79–81,139]. Meanwhile, [78] contradicted the results and identified that there is no relationship between responsiveness and customer loyalty. Furthermore, previous studies exposed that after-sale services, including return policy [140] and complaint handling [141], affects customer loyalty in a significant direct way. Moreover, [71] empirically tested the impact of delivery fulfillment and speed on customer loyalty. Thus, the MSMEs sector should improve their responsiveness level to reach a better customer loyalty level, which empirically supported RBV theory through using resources wisely to be more responsive to consumers' needs and thus increase their repurchase intention.

Finally, the illustrated positive and significant influence of product/service quality on customer loyalty was supported by [98–101,142,143]. Moreover, the impact of product quality on customer loyalty was empirically tested by [144], and the impact of service quality on customer loyalty was empirically tested by [145]. Therefore, organization, particularly MSMEs, should enhance the quality of their products/services to reach better consumers' loyalty.

Based on the previous discussion, it was concluded that the MSEs sector in emerging markets, particularly the Egyptian context, should focus on enhancing their resilience practices, namely flexibility practices and technology adoption to reach better customer loyalty level especially amongst pandemic outbreak. Additionally, they should focus on enhancing their responsiveness level either in delivery fulfillment and speed or after-sale services including handling complaints to better satisfy their customers and reach customer loyalty. Finally, customers are always seeking quality; thus, they should enhance their level of quality to reach better customer loyalty. Although the MSMEs sector has been hit by the pandemic in a dramatic way; thus, they should work harder in enhancing their resilience, responsiveness, and quality to keep their customers loyal.

## 6. Conclusions and Recommendations for Future Studies

### 6.1. Conclusions

This research aims to investigate the impact of resilience in terms of flexibility and technology adoption, responsiveness in terms of delivery speed and fulfilment and after-sale services, and product/service quality on customer loyalty in terms of the behavioral and attitudinal loyalty dimension for both manufacturing and service sectors in micro, small, and medium-sized enterprises in the Egyptian context. Based on review of the literature and previous studies, the research model was developed to conceptualize the theoretical concepts and discover the research gap, as well as the theories of RBV, DCV, and TCV through empirical evidence. The results of collected data confirmed the positive and significant relationship between resilience, responsiveness, and product/service quality and customer loyalty.

Previous studies highlighted the importance of operational resilience on customer loyalty, which was conducted separately, for example [146,147]. Moreover, the impact of responsiveness [148] and quality [149] on customer loyalty was previously conducted

separately; however, this research integrated all operational determinants on customer loyalty, where customers are the main assets for organizational survival, especially in the MSMEs sector [150].

Thus, this research collaborated the three significant variables: resilience (coping with unstable environment), responsiveness (becoming faster in fulfilling consumers' requirements), and product/service quality (enhancing quality of provided product/service) and empirically tested their effect on customer loyalty as customers are considered the main engine for organizational survival and particularly the MSMEs sector.

The authors conceptualized the proposed model of the MSMEs sector for manufacturing and service sectors from other developing countries [151,152]. Additionally, previous studies call for future studies, as suggested by [153,154]. This research will guide MSMEs' owners and practitioners to identify the critical operational performance determinants, in terms of resilience, responsiveness, and quality, which will enhance customer loyalty as the main driver for MSMEs' survival.

### 6.2. Theoretical Implications

This research focused on conceptualizing a framework investigating the impact of resilience (flexibility and technology adoption), responsiveness (delivery fulfillment and speed and after-sale services), and product/service quality on customer loyalty and confirmed the hypothesized framework in the MSMEs sector in the Egyptian context.

Moreover, the proposed framework focused on integrating the three variables together to develop a hypothesized framework, which tried to fill the research gap and provide empirical investigation on the MSMEs sector in the emerging markets.

Finally, this research contributes a richer and deeper understanding of the causal relationship between resilience (flexibility, technology adoption), responsiveness (delivery fulfillment and speed, after-sale services), product/service quality, and customer loyalty (behavioral and attitudinal loyalty dimensions).

### 6.3. Practical Implications

This research sheds light on unexplored areas regarding the impact of resilience, responsiveness, and product/service quality on customer loyalty, especially after the occurrence of the COVID-19 pandemic and its unpredictable effect on the MSMEs sector in particular [155]. Additionally, this research provides empirical evidence of the important influence of resilience, responsiveness, and quality in the improving customer loyalty level. Thus, this research can help MSMEs' managers, owners, and practitioners to focus their efforts on improving the operational resilience, responsiveness, and quality to achieve a better customer loyalty level during pandemics or similar crises, especially in emerging markets such as Egypt.

Moreover, the findings suggest that the MSMEs sector can enhance its resilience to unstable events through enhancing its flexibility practices and adopting technologies to react to the dynamic changes that occur in the market and in consumer behavior. Furthermore, the MSMEs could be more responsive through providing customers with fast and accurate deliveries and focusing on after-sales activities including complaint handling. Finally, providing products or services with good quality can lead to better customer loyalty. Therefore, these implications are particularly important for survival of the MSMEs' sector and development in challenging and emerging economies, where consumers are becoming more demanding and more strict in their buying choices and decisions.

### 6.4. Limitation and Recommendations for Futher Research

This research focused on investigating the operational resilience, namely flexibility and technology adoption, where other dimensions could be added, such as agility. Moreover, it focused on investigating responsiveness, namely delivery fulfilment and speed and after-sale service and product/service quality on customer loyalty, namely behavioral and attitudinal loyalty, where the patronization and price sensitivity loyalty dimension

can be added. Furthermore, the research model was empirically tested in micro-, small-, and medium-sized enterprises for both the manufacturing and service sectors in emerging countries, particularly Egypt, where firm size and firm sector could be used as control variables in future research. Additionally, the proposed model could be carried out for other developing and developed countries to establish a more generalizable model, and other variables could be used as a mediating variable, such as customer satisfaction.

**Author Contributions:** Introduction and hypothetical development, N.A.S. and S.E.; methodology, N.A.S., S.E. and S.M.K.; empirical analysis and findings, N.A.S.; conclusion and recommendations, N.A.S.; supervision, S.E. and S.M.K.; project administration, S.E. and S.M.K. All authors have read and agreed to the published version of the manuscript.

**Funding:** This research received no external funding.

**Institutional Review Board Statement:** Not applicable.

**Informed Consent Statement:** Not applicable.

**Data Availability Statement:** Data available upon request from corresponding author.

**Conflicts of Interest:** The authors declare no conflict of interest.

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
