# Peer review of "Investigating the Impact of Resilience, Responsiveness, and Quality on Customer Loyalty of MSMEs: Empirical Evidence"

_sustainability, doi:10.3390/su14095011_

Round 1

Reviewer 1 Report

I hope that my suggestions will assist you in improving the manuscript.

INTRODUCTION

  • The authors present a good flow in justifying the needs of this study. The choice of customer loyalty as the focus of this study, as well as the selection of resilience, responsiveness and service quality, is well-justified.  

LITERATURE REVIEW

  • Section 2.1. The discussion in this section mainly focused on the resilience and organizational performance or how resilience can help organizations be sustainable and meet customer needs, with no mention of the relationship between resilience and customer loyalty. So, what is the basis for the H1 proposal?
  • Please avoid abbreviating organizational responsiveness with "OR," as it could be misinterpreted as "or."
  • Section 2.2. Hope authors can provide more discussion on the relationship between resilience and customer loyalty to justify the formulation of H2.
  • Section 2.3. There is no discussion about the relationship between service quality and customer loyalty. The argumentation only on the important of quality on operational performance and customer needs. Authors should bear in mind that rigorous arguments is a prerequisite to make sense of the proposal of hypotheses (H1-H3). Please improve the sections accordingly.
  • The RBV, DVC, and TCV underpinning theories were briefly mentioned in the introduction section. However, no further explanation is given as to how these three theories can be integrated and used as the theoretical foundation for the hypothesized relationships depicted in Figure 1.

METHODOLOGY

  • Section 3.1, line 244. Authors mentioned that 240 respondent replied and got 9 incomplete responses. But, why the valid Responses is 224 and not 231? Please clarify.
  • Section 3.3. Please double-check whether 224 or 231 is the correct valid data for the pilot test.
  • Section 4.2. Using Fornell and Larcker criterion to assess discriminant validity had received lot of criticism from statistical scholars. I would suggest using heterotrait-monotrait ratio of correlations (HTMT) – a more valid assessment for discriminant validity that recommended by Henseler, Ringle and Sarstedt (2015).

FINDINGS

  • Please include a section named "discussion" to discuss your findings. Compare your findings to previous studies and explain "why" such a result was obtained in your research context.

IMPLICATIONS

The practical implications are discussed. The theoretical implication section, however, has been omitted by the authors. So, what is the new contribution of this study in comparison to previous research works?

Author Response

Dear Reviewer

Thank you so much for the time and effort that you have dedicated for providing your valuable feedback on our manuscript and thanks for giving us the opportunity to re-submit a revised manuscript. 

Kindly find the attachement

Reviewer 2 Report

Thank you for sending this paper to review. I have read the entire article with great interest. Might the below point help the authors to improve their manuscript.

  1. In title, authors used SMEs, while in entire manuscript they spoke about MSMEs. Please rectify this issue.
  2. In abstract, authors must briefly mention the negative impact of the covid-19 pandemic on MSMEs to justify the problem. Also, the full form of RBV, DCV, and TCV must be mentioned.
  3. Introduction must start with the problem behind the investigation, and impact of covid-19 as it is the main focus mentioned in the abstract. There is lack of connection between paragraphs in the introduction. Please reorganize the introduction.
  4. The research gap and novelty of the research should be discussed in the introduction.
  5. In abstract, authors mentioned to extend the RBV, DCV, and TCV, but in introduction line 91-92 only RBV and DCV is focused. Please rectify this issue.
  6. Line 102 to 107 seems like a fragmented and lengthy statement.
  7. In literature, authors fail to provide a debate of studies that considered the examined variables. The literature section is fragmented and lack of connection between two paragraphs. Authors are suggested to rewrite or reorganize it. So many definitions for the meaning of OR seem ambiguous.
  8. Authors must provide references and findings of previous studies in different scenarios to justify the proposed hypothesis. In current form, literature review is just the meaning of different variables and not providing any review of previous studies which accepted or denied the association of two variables. It is suggested to the authors to review other articles to organize your literature section. You can follow this article for instance-

Raza, A., Saeed, A., Iqbal, M. K., Saeed, U., Sadiq, I., & Faraz, N. A. (2020). Linking corporate social responsibility to customer loyalty through co-creation and customer company identification: Exploring sequential mediation mechanism. Sustainability, 12(6), 2525.

  1. Another major aim of this study (as presented in the abstract) is to extend the theories of RBV, DCV, and TCV. However, authors did not consider it in the literature section, like what this theory refers to, and what is the need of extending these theories. Furthermore, there is no argument of these theories in results and discussion section. It is not clear how authors extended these theories. Just mentioning theories in abstract and conclusion does not fulfill the objective of the research.
  2. In material and methods, please mention the version of SPSS and AMOS used for the study.
  3. In line 228 and 230, SPSS and AMOS were not conducted but used for the analysis. Please correct the statement.
  4. In section 3.1, authors must describe the adequacy of the sample size used for the study through a reference for minimum sample size for conducting CFA and CB SEM and brief reason to justify the usage of snowball sampling.
  5. Section 3.2 line 253 is incomplete.
  6. Line 275, what is the significance of citation [92, 93] in the middle of sentence, as the statement is not providing any relevant information.
  7. In section 3.3, authors must mention the items removed because of low factor loadings.
  8. Section 4, authors must justify why CB SEM is used for analysis and why it is better than PLS SEM for this study. Provide relevant references.
  9. Section 4.2, give citations for threshold values.
  10. The explanation of discriminant validity is not clear. What you compared with square of AVE and what results you got? Please describe. Also, it is suggested to use both Fornell and Larcker criteria and HTMT to assess and describe discriminant validity.
  11. Authors did not provide the threshold values of model fit indices anywhere in the study. Provide relevant references suggesting threshold values for each assessed model fit indices. Also, the value of (x2/df = 5.965) in your study is more than 5, how do you justify it. Please provide some references as many scholars recommend its value to be 5 or less. In addition, RMSEA is a critical index in SEM, please assess and provide its value.
  12. Authors did not perform any test for justifying multi-collinearity issue. It is suggested to provide variance inflation factor (VIF) values for each variable.
  13. Discussion of the findings is not sufficient. Authors must discuss the novel results of their study and significance behind it and how these results are helpful. Just mentioning what other studies found is not satisfactory.
  14. Conclusion and practical implications must  be improved.
  15. There are many spelling mistakes in entire manuscript. Also, it is suggested to get the manuscript proofread by a native English speaker.

Wish you good luck.

Author Response

Dear Reviewer

Thank you so much for the time and effort that you have dedicated for providing your valuable feedback on our manuscript and thanks for giving us the opportunity to re-submit a revised manuscript. 

Kindly find the attachment

Round 2

Reviewer 1 Report

Thank you for the revised version. There are few issues that need your further attention and improvement. Just a gentle reminder, in your next submission, please ensure the section and line number in the response table are correctly stated to ease the review process.

LITERATURE REVIEW

  • Section 2.2. I noticed the authors included a brief summary of previous studies on the relationship between organizational responsiveness and customer loyalty in a short paragraph. However, the justification for the formulation of H2 remains unclear as to "how" and "why" this variable influences customer loyalty.
  • Section 2.3. The same goes to this section. The formulation of H3 is also unjustified. The information added only a summary of pass studies; the discussion – ‘how’ product quality influences customer loyalty was insufficient.
  • Page 6, Figure 1. What have been added only a brief mention of the theories (RBV, DVC, and TCV). What are their underlying assumptions? Which relationships did they underpin? Where is the argumentation about ‘how’ these theories can be integrated to explain the hypothesized relationships?

DISCUSSION

The authors compare their findings to previous studies, and discuss what MSMEs can do. They do not, however, explain "why" and “how” resilience, responsiveness and quality can influence customer loyalty in their research context.

IMPLICATIONS

  • The theoretical implication section has been added, but the argumentation was inconclusive. To justify the theoretical implications, it is unnecessary to claim that this is the first/no study. To claim that this is the first study, the authors must conduct a thorough literature review and provide sufficient evidence to convice the readers.

Author Response

Dear Reviewer,

Thank you for all the comments that helps us to improve quality of the paper 

Kindly find the attached file

Reviewer 2 Report

The quality of manuscript is enhanced after review. I accept the changes made by the authors.

However, Spell check is required for entire manuscript.

Author Response

Dear Reviewer,

Thank you for all your comments throughout the review process that improved the quality of paper

Kindly find the attached file
